# Temporal markers in a temperate ice core: insights from $^{3}$H and $^{137}$Cs profiles in the Adamello glacier

Elena Di Stefano[1,2], Giovanni Baccolo[3], Massimiliano Clemenza[2], Barbara Delmonte[1], Deborah Fiorini[1], Roberto Garzonio[1], Margit Schwikowski[3], and Valter Maggi[1]

[1]Environmental and Earth Sciences Department, University Milano-Bicocca, 20126 Milano, Italy
[2]Milano-Bicocca Section, Istituto Nazionale di Fisica Nucleare, 20126 Milano, Italy
[3]Paul Scherrer Institute, 5232 Villigen PSI, Switzerland
**Correspondence:** Elena Di Stefano (elena.distefano@unimib.it)

**Abstract.** The article discusses the use of tritium ($^{3}$H) and cesium ($^{137}$Cs) as temporal markers in ice cores extracted from temperate glaciers. We present a complete tritium profile for a 46 m ice core drilled from Adamello Glacier, a temperate glacier in the Italian Alps, and compare it to the $^{137}$Cs profile from the same ice core. Our analysis reveals tritium contamination between 19 and 32 m of depth, which can be attributed to the global radioactive fallout caused by atmospheric nuclear bomb testing in the 1950s and 1960s. Results show that the radioactive peak linked with 1963 does not occur at the same depth for both $^{3}$H and $^{137}$Cs, but the tritium peak is 1.5 m above the cesium one. Our hypothesis is that this misalignment is caused by meltwater-induced postdepositional processes that affect $^{137}$Cs, which is more sensitive to percolation than $^{3}$H. The total inventory of $^{137}$Cs in this ice core is also among the lowest ever reported, providing additional evidence that the presence of meltwater affected the distribution of this radionuclide inside the ice. On the contrary, the total tritium inventory is comparable to what is reported in the literature, making it a more reliable temporal marker for temperate glaciers.

## 1 Introduction

Due to its sensitivity to temperature variations, the cryosphere offers some of the most notable evidence of climate change, with dramatic ice losses reported worldwide. In particular, mountain glaciers are globally losing mass at accelerating rates (Zemp et al., 2019) and according to future projections, including the most optimistic ones, this trend will continue throughout the current century (Hock et al., 2019). European alpine glaciers are amongst the fastest declining ones, with an estimated average annual ice loss of -0.74 ± 0.20 meters water equivalent for the period 2000-2016 (Davaze et al., 2020). As a result of increasing temperatures, glaciers are not only retreating, they are also undergoing changes in their thermal regime. Atmospheric warming and the increasing number of melting events are rapidly modifying the temperature distribution of glaciers, increasing ice temperature (Gilbert et al., 2010, 2020). The occurrence of accumulation basins with temperature consistently below the pressure melting point is increasingly rare in many mountain ranges, as they transition from cold to temperate conditions (Gabrielli et al., 2010). Temperate ice is defined as ice that can contain meltwater inclusions due to its thermodynamic state (Lliboutry, 1971). Moreover, the rise of the Equilibrium Line Altitude (ELA), defined here as the altitude at which a glacier has a zero mass balance, is leading to a gradual reduction in glacier size, ultimately leading to the disappereance of accumulation

basins. It is expected that by the end of the 21st century, at least 69% of the glaciers in the Alps will be entirely below the ELA (Zebre et al., 2021). As a consequence of increased temperature and subsequent percolation of meltwater, it is becoming increasingly difficult to retrieve reliable climatic records from mountain glaciers through ice core drilling. In the future, the role of glaciers as paleoclimatic and environmental archives will depend on our ability to interpret signals from temperate glaciers. Determining if and to what degree such glaciers can provide climatic and environmental records is an urgent issue that ice core science must address. Meltwater percolation poses a potential threat to the preservation of proxy signals in the ice. Possible effects include smoothing the temperature isotopic signal (Thompson et al., 1993), relocating impurities (Pavlova et al., 2015) and altering trace elements records. Because meltwater-induced postdepositional effects vary among chemical species and can be influenced by several factors (Avak et al., 2018; Moser et al., 2023) including solubility, the position inside the ice lattice, and the concentration, it can be sometimes difficult to evaluate to what degree the signal has been altered. Generally, elements and compounds that are insoluble in water, present in particle form, or well incorporated into the ice lattice, will be less prone to elution and relocation due to meltwater (Eichler et al., 2001; Wong et al., 2013). Microparticles may still accumulate at melt surfaces, leading to the enrichment of elements bound to particulate matter (Pavlova et al., 2015; Niu et al., 2017).

Among the many contaminants we can find in ice cores there are radionuclides of anthropogenic origin. During the 1950s and 1960s, atmospheric nuclear bomb testing carried out mainly by the USA and USSR resulted in the global radioactive contamination of the environment. Atmospheric thermonuclear explosions can reach the stratosphere, where the circulation allows the contamination to rapidly extend to the whole hemisphere. The majority of tests were carried out in the northern hemisphere, which therefore suffers from the highest contamination, although radionuclides generated from atmospheric nuclear tests have been found in the Antarctic continent as well (Picciotto and Wilgain, 1963; Jouzel et al., 1979). In 1963 the atmospheric yield of nuclear bomb testing reached its maximum (United Nations Scientific Committee on the Effects of Atomic Radiation, 2000), before the entering into force of the Partial Test Ban Treaty, which introduced a ban on atmospheric nuclear testing for USA, USSR, and UK. This maximum has been recorded in paleoclimatic archives and has subsequently been used as a temporal reference horizon (Mikhalenko et al., 2015; Eichler et al., 2000; Clemenza et al., 2012). Likewise, radioactive layers relative to the 1986 Chernobyl accident (United Nations Scientific Committee on the Effects of Atomic Radiation, 2008; International Atomic Energy Agency, 2006) and the 2011 Fukushima accident (Steinhauser et al., 2014) have been found in environmental records. The main radionuclides injected into the environment following such events include cesium ($^{137}$Cs) and tritium ($^3$H), both of which have been extensively documented in glaciers and ice caps. The radioactive layers corresponding to 1963 and 1986 can aid the dating of ice cores, if the signal is well preserved, and can shed some light on melting processes inside the glacier (Kang et al., 2015).

Here we present the complete $^3$H profile for a 46 m ice core extracted from the Adamello glacier, a low-altitude temperate glacier. To assess the integrity of signals in temperate ice, the tritium record is also compared with $^{137}$Cs and hyperspectral data related to dust content. Finally, a comparison is made with similar data from a cold glacier (Colle del Lys).

## 1.1 Tritium

Tritium is a pure beta emitter with a maximum emission energy of 18 keV and a half-life of 12.33 years (Lucas and Unterweger, 2000). It is naturally produced in the upper atmosphere as a result of the interaction between cosmic rays and atmospheric gases, mainly nitrogen, at a production rate of 0.320 atoms cm$^{-2}$ s$^{-1}$ (Masarik and Beer, 2009). This results in a low natural background detectable in precipitations estimated to be <1 Bq L$^{-1}$ in the United States (Thatcher, 1962; Kaufman and Libby, 1954), although pre-1950s measurements, when the artificial production of tritium began, are quite scarce. Reconstructions of the natural tritium background are available from ice core data from Fiescherhorn glacier (Schotterer et al., 1998a).

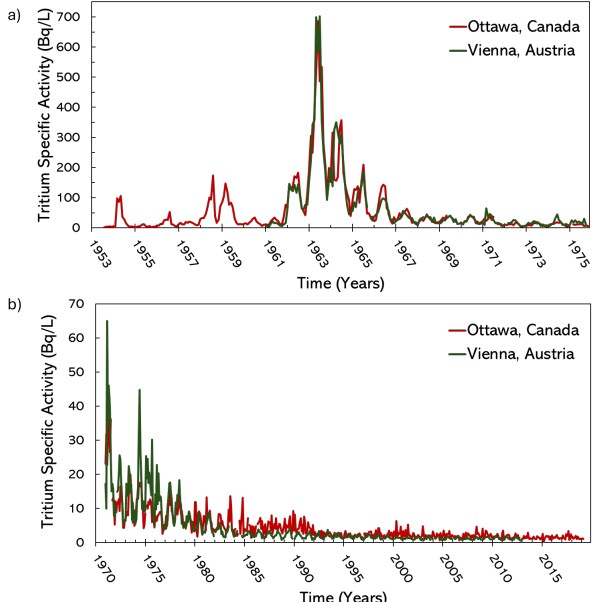

**Figure 1.** Monthly concentration of tritium in precipitation in Vienna (Austria) and Ottawa (Canada). Panel (a) shows data from 1953 to 1975, while panel (b) shows data from 1971 to 2019. This data was obtained from the Global Network of Isotopes in Precipitation (GNIP) Database (International Atomic Energy Agency and World Meteorological Organization, 2021) and converted to Bq L$^{-1}$ (where 1 Tritium Unit = 0.118 Bq L$^{-1}$). Please note the different scale of Y-Axes in the two panels.

$^{3}$H was a key component in thermonuclear bombs, being one of the hydrogen isotopes used as thermonuclear fuel. During the nuclear testing period, tritium concentrations in Northern Hemisphere precipitations reached several thousands Tritium Units (TU). Figure 1 shows monthly tritium concentration in precipitation for Vienna and Ottawa, obtained from the Global Network of Isotopes in Precipitation (GNIP) Database (International Atomic Energy Agency and World Meteorological Organization, 2021). These trends, representative of the Northern Hemisphere, show a gradual increase during the 1950s leading up to a maximum in 1963, followed by a gradual decrease after the enactment of the Partial Test Ban Treaty. Tritium activity concentration show a strong seasonality, with peak values occurring in spring-summer each year. This phenomenon, known as

'Spring Leak' (Michel, 2005; Harms et al., 2016), is due to enhanced air exchange between the troposphere and the stratosphere, the main reservoir for this radionuclide, during spring. [3]H has been extensively used as a temporal horizon marker for 1963 in ice cores (Eichler et al., 2000; Qiao et al., 2021; Mikhalenko et al., 2015; Gabrieli et al., 2011; Kang et al., 2015) and, as long as activities were high, to reveal seasonal signals and aid annual layer counting in ice cores (Schwikowski et al., 1999; Schotterer et al., 1998b; Yasunari et al., 2007). Given tritium's relatively short half-life, its detection is gradually becoming

more challenging as more time passes since its emission into the environment.

## 1.2 Adamello Ice core

The ADA16 ice core was drilled during the spring of 2016 at Pian di Neve (WGS84: 10°31'22" E, 46°8'51" N), the former accumulation basin for the Adamello Glacier; this glacier has experienced several years of negative mass balance (Ranzi et al., 2010) and is now entirely located below the ELA. The first chronology for this ice core has estimated a surface age of 1993,

indicating no accumulation was preserved on this glacier for the last 20 years (Festi et al., 2021). Due to its relatively low altitude (3100 m asl) and prevailing climatic conditions, Adamello glacier has a temperate regime: the body of the glacier is at melting point and during summer is massively subject to melting, which could alter the climatic signal preserved in the ice. The ice core was drilled where the maximum ice thickness of the glacier was estimated (268 $\pm$ 5 m) (Picotti et al., 2017) and reached a final depth of 46 m. During drilling, ice core chips were also collected. These are residual materials produced during

the mechanical drilling by the contact between the ice and the rotating blade and are usually discarded due to potential chemical contamination, as the ice has been in direct contact with the drill. Nonetheless, they are suitable for radionuclide analysis, as radioactive contamination during drilling operations is virtually impossible, although they have a poor depth resolution which is limited to the entire length of the single ice core sections extracted during drilling runs. The chips were collected in a specific close chamber above the core barrel, which was emptied after the collection of each core sample.

Analyses were performed both on the chips and on ice samples from the ice core. The chips were divided into 112 runs, with some stored together, for a total of 71 samples covering each a mean thickness of 59.4 cm. Ice samples were taken from 27 m to 35 m of depth to obtain a higher temporal resolution for this part of the ice core, with samples collected at intervals of five to ten centimeters.

## 1.3 Lys Ice core

To have a reference from a cold alpine glacier, we also analyzed 38 samples belonging to an ice core drilled at Colle del Lys (7°51'4" E, 45°55'13" N), which is the uppermost part of the Lys glacier, located in the Monte Rosa massif at 4240 m asl. This is one of the few glacierized areas presenting cold ice in the Alps: thanks to their elevation, glaciers in the upper regions of the Monte Rosa massif are only occasionally subjected to melting (Hoelzle et al., 2011). Consequently, several undisturbed ice cores were retrieved from this area, specifically from Colle del Lys and Colle Gnifetti (Smiraglia et al., 2000; Villa et al.,

2006; Wagenbach et al., 2012). For the purposes of this study, we analyzed samples remaining from the [137]Cs analysis that was carried out by Clemenza et al. (2012).

## 2 Materials and Methods

### 2.1 Sample Preparation and Analysis

Samples from ADA16 were thawed in a clean room inside precleaned polyethylene bottles and then filtered using a vacuum-driven filtration system and polycarbonate filters (47 mm diameter, 0.45 µm cutoff), then passed through Eichrom's Tritium Column, to eliminate all possible radioisotopic disturbances present in the sample. Tritium columns were first conditioned using ten ml of ultrapure water, before adding 25 ml of sample, of which the first five ml were discarded while the rest was collected for further preparation. At last, samples were prepared for liquid scintillation counting by mixing 8 ml of sample with 12 ml of scintillation cocktail (UltimaGold ullT) in a 20 ml polypropylene low diffusion vial.

Samples from Lys Glacier were directly prepared for liquid scintillation and measured using a total beta protocol. Data from this ice core will be presented as counts per minute (cpm) due to the inability to construct a calibration curve for total beta analysis.

The liquid scintillation counter used is the Quantulus 1220, manufactured by Perkin Elmer. It is equipped with two dual multi-channel analyzers, enabling simultaneous measurement of four spectra, each with 1024 channel resolution. In our analysis, we considered a counting window between 50 and 250, which covers the emission energy of tritium.

### 2.2 Calibration curves and Detection Limit

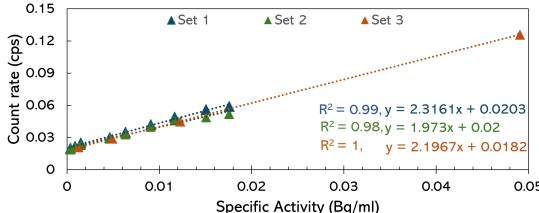

**Figure 2.** Calibration curves calculated for the three sets of tritium standards. Data points are plotted as counts per second (cps) vs activity values (Bq/ml). For each set the dashed line represents the linear fit between the data points, which is also reported in the left corner of the image together with the coefficient of determination ($R^2$). Errors on the count rate, calculated as the square root of each measurement divided by the count time, are 0.001 cps.

For conversion from the count rate (cps) to the Specific Activity ($Bq\,L^{-1}$) we constructed a calibration curve for each dataset (low-resolution chips and high-resolution ice core) by measuring tritium standards of known activity, as can be seen in Figure 2. Standards were prepared from NIST Hydrogen-3 Radioactivity Standard 4361C. Three sets of standards (Set1-2-3) were prepared following the same procedure that was used for the samples, the only exception being that Set2 was the only one that passed through tritium columns. Set1 and Set2 were prepared with identical activities to estimate the potential influence of the elution columns on sample activities. As the difference between Set1 and Set2 was below 15% (in terms of cps), we decided

not to elute the tritium standards. Set3 was repeatedly measured in between sample runs to check for instrumental stability. Set1 was used for calibration of the low-resolution dataset, while Set3 was employed for the high-resolution dataset. The errors

for the count rate were calculated as the square root of each measurement divided by the count time.

Following Currie (1968) we calculated a Critical Level as $LC = 1.64\sqrt{2Rb/t}$ where Rb is the background rate and t is the measurement time; when the net sample rate is higher than the critical level, we can consider the sample contribution (Rs-Rb) to be distinct from the background, and the Specific Activity (Bq L$^{-1}$) was calculated using the calibration coefficient from the standard measurements. All activities were decay corrected to 01 January 2016 and the associated uncertainty was calculated

employing the propagation of error method. When (Rs-Rb) < LC, an upper limit was calculated as three times the counting error on the sample measurement. The Detection Limit for our measurement is 1.5 Bq L$^{-1}$, calculated as 3.3 $\sigma$ where $\sigma$ is the standard deviation of ten independent blanks (Currie, 1968).

## 2.3  Hyperspectral imaging spectroscopy

Hyperspectral imaging spectroscopy is a technique used to characterize surfaces and materials on the basis of their optical

properties. In the visible part of the electromagnetic spectrum (from 300 to 750 nm), snow and ice show high reflectance; ice core characteristics such as the impurity content (e.g., mineral dust, volcanic ash, black carbon, algae) affect the ice's capability to absorb and reflect electromagnetic radiation in the visible range, allowing the extraction of a continuous record of reflectance and derived parameters along the ice core. Ice core sections were placed on a fixed ice-core support, illuminated using a halogen stable light source (600 or 1000 W, LOT Quantum Design), and scanned with a HeadWall spectrometer (400

140  - 1000 nm, Hyperspec VNIR, HeadWall Photonics). The Hyperspectral imaging scanner was set in motion step by step from the top to the bottom of sections, using a motion speed of 0.9 mm s$^{-1}$ and an exposure time per frame of 39.8 ms. The high-resolution images were then processed to obtain reflectance curves focusing on a 20-pixel section centered along the central axis of ice core sections. The technique, developed in collaboration between the European Cold Laboratory (EuroCold) and the Remote Sensing of Environmental Dynamics Laboratory (LTDA) both at University of Milano-Bicocca, is described in

detail in Garzonio et al. (2018). For this study, we analyzed a portion of the ice core about seven meters long (from 26.8 m to 33.9 m), corresponding to the area around the main $^{137}$Cs peak. Our objective was to investigate whether the presence of impurities could impact the distribution of radionuclides in the ice, given previous research demonstrating a significant association between cesium and particulate matter (Di Stefano et al., 2019). Finally, we calculated the Snow Darkening Index (SDI) as the normalized ratio between the reflectance in the red (i.e., from 640 to 670 nm) and green (i.e., from 550 to 590 nm)

wavelengths, providing information on the concentration of mineral dust in the ice core (Di Mauro et al., 2015).

## 3   Results and Discussion

### 3.1   ³H Activity

The tritium profile for the Adamello ice core is presented in Figure 3. The low-resolution dataset covered the whole depth of the ice core, but only the portion between 19 and 32 m displayed a tritium signal, implying these layers can be attributed to the period between the 1950s and the 1970s.

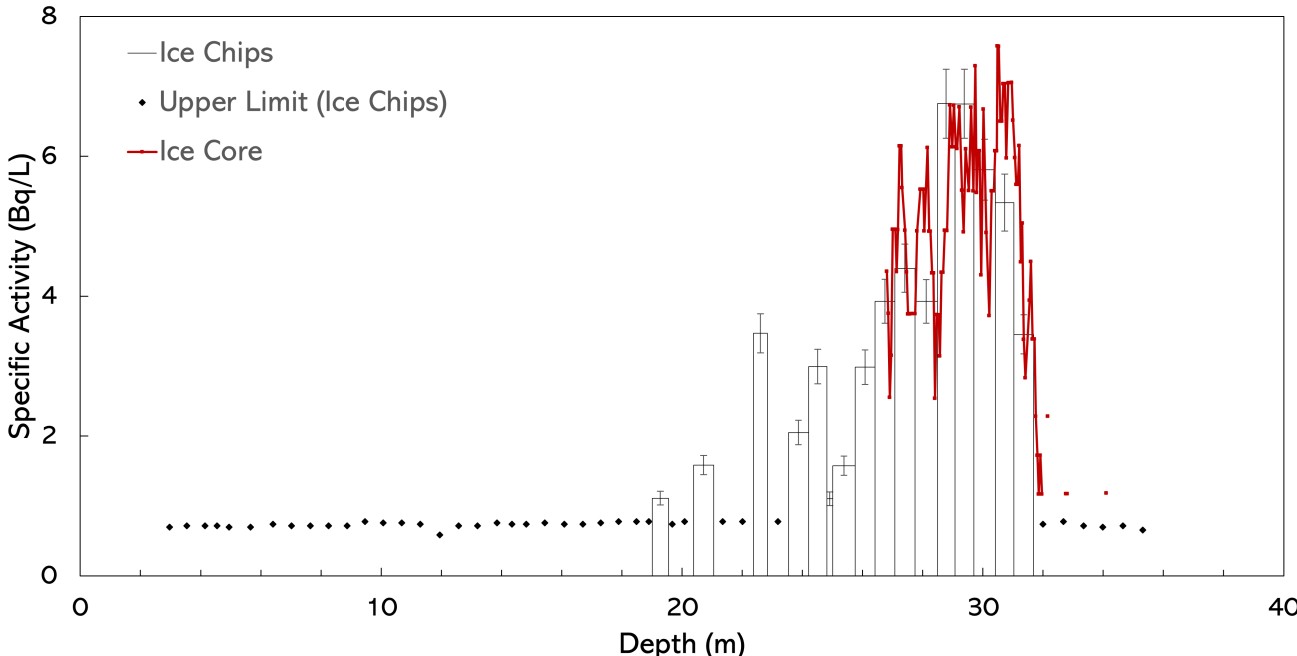

**Figure 3.** Tritium activity in the Adamello temperate ice core. Low-resolution measurements referring to ice chips are shown as black histograms where the width of each bar represents the actual depth covered by the sample and the height represents the Specific Activity (Bq/ml). Error bars indicate the uncertainty on the Specific Activity, calculated following the propagation of error method. Diamonds represent samples from the low-resolution dataset for which the Upper Limit was calculated instead of the Specific Activity, as detailed in the main text. High resolution measurements referring to ice core samples are shown in red.

The tritium signal was further investigated by analyzing the high-resolution dataset between 27 and 35 m. In general, good accordance between the two datasets was found, both regarding the measured Specific Activities and the general trend, with the exception of the main peak which is found at $29 \pm 0.6$ m in the low-resolution dataset and at $30.530 \pm 0.025$ m for the high-resolution dataset. We consider the low-resolution dataset to be less reliable due to its broader depth coverage which may lead to inaccurancies when the signal exhibits peaks and valleys compressed in a narrow depth range, as shown by the high-resolution dataset. Activity values are generally lower than those reported for other alpine glaciers (Eichler et al., 2000; Schotterer et al.,

1998b, 1977; Oerter and Rauert, 1982) even when accouting for decay. Discrepancies between tritium activities as reported in precipitation at the time of the radioactive fallout and tritium detected in ice cores can be explained by i) immediate loss of a fraction of the initial deposited signal due to summer snow melt, as proposed by Oerter and Rauert (1982); ii)current loss due to melting and water percolation inside the ice, which is expected to be more extensive in the Adamello ice core than in ice cores extracted from cold glaciers.

## 3.2 $^{137}$Cs and $^3$H profile comparison

It is interesting to compare the tritium signal with the $^{137}$Cs profile from the same ice core, as reported by Di Stefano et al. (2019). $^{137}$Cs is also a byproduct of nuclear atmospheric tests, and since the transport mechanisms of these two radionuclides are similar (Pourchet and Pinglot, 1979), it is expected to find the 1963 peak at the same depth for $^{137}$Cs and $^3$H, as observed in ice cores from cold glaciers (Pinglot et al., 2003). We note that in this study, the low-resolution tritium dataset was prepared from the same exact samples used for $^{137}$Cs analysis.

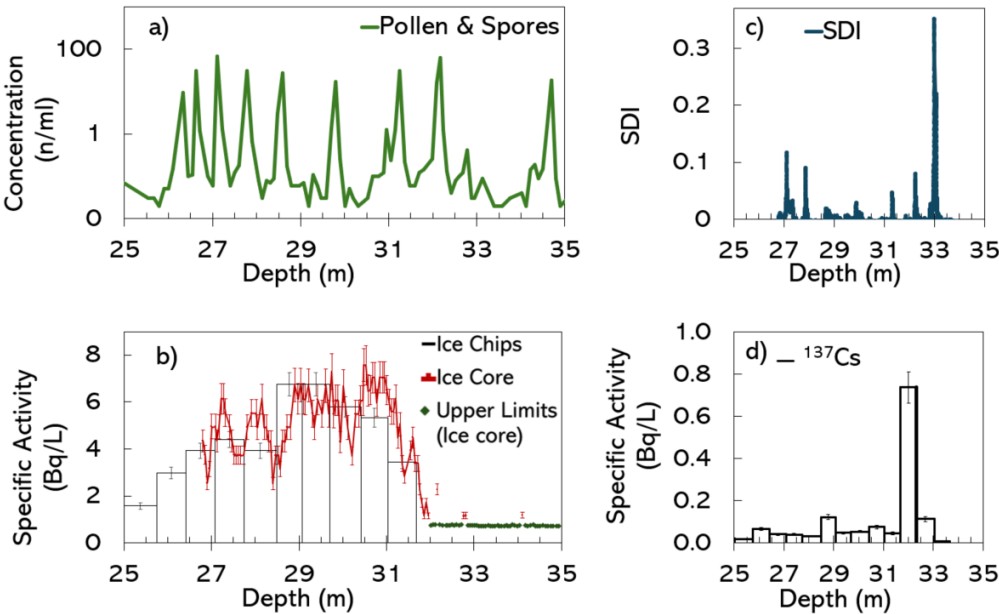

**Figure 4.** Detailed overview of the ice core in the portion where the the tritium and $^{137}$Cs 1963 peaks have been observed (between 25 and 35 m deep). a) Pollen and spores concentration taken from Festi et al. (2021). Each peak of pollen and spores concentration reflects one flowering year (February–September). b) Tritium profiles. Specific activity (Bq L$^{-1}$) of $^3$H for the ice chip low-resolution dataset is shown as black histograms while the ice core high-resolution dataset is shown in red. Error bars indicate the uncertainty on the Specific Activity, calculated following the propagation of error method. The diamonds indicate upper limits calculated for the high-resolution data set, as detailed in the main text. c)Snow Darkening Index (SDI) record, a hyperspectral index related to the impurity content of ice. d) Specific activity (Bq L$^{-1}$) of $^{137}$Cs taken from Di Stefano et al. (2019)

In the Adamello ice core, the respective peaks are misaligned by at least 1.5 m when considering the high-resolution dataset for tritium, as can be seen in Figure 4 (b,d). Based on the dating proposed by Festi et al. (2021), this would correspond to a time lag of approximately two years between the two peaks, which in terms of deposition would be difficult to explain. Our hypothesis is thus that the $^{137}$Cs signal has been altered by post-depositional processes. We must note that Festi et al. (2021) used data from Di Stefano et al. (2019) as a tie point for 1963, therefore the established timescale may be slightly inaccurate regarding the layer counting between the 1986 and 1963 tie points, as it heavily relies on the assumption that the $^{137}$Cs peak corresponded to the 1963 unaltered signal. This inaccuracy would introduce an additional error of a couple of years.

In Figure 4c, the SDI data shows a distinct impurity peak at 33 m of depth, indicating a layer rich in mineral dust. This layer, found in section 77-78, is approximately one meter deeper than the main $^{137}$Cs peak (section 74-75). The presence of a significant impurity layer beneath the $^{137}$Cs peak suggests that the latter is not the result of dust enrichment in a single layer as a result of melting. While impurities can accumulate in a single layer if there is a hidden melting surface within the glacier, potentially leading to radionuclide concentration (Baccolo et al., 2020), the absence of any significant peak in the SDI dataset at the same depth as the main cesium peak indicates that the cesium peak is not caused by an abundance of dust in this layer. This finding is significant because the cesium data is not calibrated on the amount of dust present but on the volume of filtered water, despite $^{137}$Cs being in particulate form. Moreover, the absence of a dust-rich layer at 32 m of depth suggests that even a minimal amount of particulate matter can contribute to a substantial cesium peak.

Our proposed explanation for the lag between the $^3$H and the $^{137}$Cs signals is the presence of meltwater-induced postdepositional processes altering the signal embedded in the ice. $^{137}$Cs, mainly bound to particulate matter (Qin et al., 2012; Tanaka et al., 2013; Di Stefano et al., 2019), is more susceptible to percolation than $^3$H (Pinglot et al., 2003), which is incorporated in water molecules and considered a matrix signal. This result was unexpected, as particle-bound signals are generally well preserved within this ice core, leading us to expect similar behavior from cesium. In fact, particulate matter impurities are usually among the last proxies to be affected by meltwater relocation processes (Meyer and Wania, 2011; Moser et al., 2023). The exact mechanism behind the relocation of cesium is beyond the scope of this study. Further research is needed to investigate the relationship between cesium and the presence of meltwater in ice cores.

Vertical relocation of cesium has been reported in the literature: Pinglot et al. (2003) found a similar delay between cesium and tritium in ice cores extracted from low altitude glaciers in Svalbard, although the difference between the $^3$H peak and the $^{137}$Cs was less pronounced (<1 m). This can be expected since Adamello glacier has experienced much more melting than Svalbard and more time has passed since the deposition of radioactive species on the glacier. Similarly, Schotterer et al. (1977) reported a high concentration of $^{137}$Cs three meters below the tritium bomb level in an ice core extracted from a relatively low altitude glacier in the Swiss Alps (Plaine Morte).

Our hypothesis is further supported by findings we obtained on total beta measurements from an ice core extracted at Colle del Lys in 1996 (Smiraglia et al., 2000). Figure 5 shows that in this ice core the main beta ($^3$H) peak and the main $^{137}$Cs coincide. Due to its position and high elevation, Lys glacier is characterized by a cold regime, as proven by the borehole temperature measurements taken at the time of drilling (Smiraglia et al., 2000). This allows the radioactivity signal to be preserved in the ice without any relocation. This finding is consistent with another study conducted on an alpine cold glacier, Grenzgletscher,

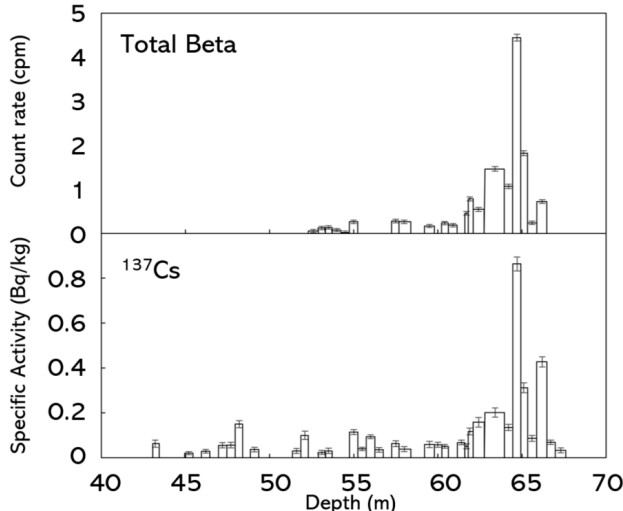

**Figure 5.** The upper panel shows the total beta activity from the Lys ice core. Total beta values are reported here as count rate in counts per minute (cpm) with the associated uncertainty. The lower panel shows the Specific Activity (Bq/kg) of $^{137}$Cs with the associated error, as reported in Clemenza et al. (2012). In both panels the width of each bar represents the actual depth covered by each sample.

where the 1963 peak for beta/tritium and cesium were observed at the same depth (Eichler et al., 2000). Grenzgletscher and Lys drilling sites are less than 15 km apart.

## 3.3  $^{137}$Cs and $^3$H total inventories

For the Adamello ice core, we calculated the integrated activity of both $^{137}$Cs and $^3$H related to the 1963 fallout events. To do so, data in Bq L$^{-1}$ was converted to Bq m$^{-2}$, considering the thickness of each sample and the ice density of 0.9 g cm$^{-3}$. This data was also decay corrected to 1963. Regarding $^{137}$Cs, acknowledging the signal migration, and in order to be conservative in our calculations, we considered everything below 28 m of depth to be possibly related to the nuclear weapons fallout; taking this into account we obtained a total inventory of $^{137}$Cs of $2597 \pm 400$ Bq m$^{-2}$, where the main contribution is given by a single sample ($1519 \pm 228$ Bq m$^{-2}$). This value is at the lower limit with respect to what observed in other alpine glaciers ($3500 \pm 800$ Bq m$^{-2}$)(Eichler et al., 2000) and in European soils (e.g. Bossew et al. (2001) reports a value of $^{137}$Cs from bomb fallout of 2300 Bq m$^{-2}$ in 1986, which corresponds to approximately 3800 Bq m$^{-2}$ if decay corrected to 1963). These results indicate partial signal loss due to meltwater drainage for the $^{137}$Cs signal. Looking at the high-resolution tritium dataset, we calculated a total inventory of $71000 \pm 7000$ Bq m$^{-2}$ for the 1963 event (considering from 30,5 to 31 m), corrected to 1963, with the peak at 30.5 m corresponding to $7300 \pm 700$ Bq m$^{-2}$. For comparison, Oerter and Rauert (1982) in a study of two ice cores extracted from Vernagtferner Glacier (Austrain Alps), sampled at a similar thickness resolution than ours, found the highest tritium values, corrected to 1963, to be 4500 and 5300 Bq m$^{-2}$. In our low-resolution dataset, the single highest sample corresponds to $75000 \pm 5500$ Bq m$^{-2}$. To roughly compare our data to literature, we converted data reported in literature to

225 Bq m$^{-2}$ and decay corrected it to 1963, as seen in Table 1. Our data is comparable if not higher than reported tritium activities in other ice cores, suggesting that tritium in the Adamello ice core was not lost due to melting, although the region around the main peak appears to be broader than what seen in literature, resulting in activity values in Bq L$^{-1}$ being lower than expected, likely due to diffusion processes.

**Table 1.** Activity values for the 1963 tritium peak at different locations. Data taken from literature were converted to Bq m$^{-2}$ considering an ice density of 0.9 g cm$^{-3}$ and decay corrected to 1963, but are to be considered rough approximations. Reported data refers to the single highest sample if not stated otherwise.

| Ice Core Location | Reference | 1963 Tritium Inventory (Bq m$^{-2}$) |
|---|---|---|
| Geladaindong, Tibetan Plateau | Kang et al. (2015) | 84000* |
| Grenzglestcher, Swiss Alps | Eichler et al. (2000) | 25700 |
| Jungfraujoch, Swiss Alps | Schotterer et al. (1977) | 47800 |
| Adamello, Italian Alps | This work (Ice core) | 71000** |
| Adamello, Italian Alps | This work (Ice chips) | 75000 |

*Sum of 3 different datapoints

**Sum of all datapoints between 30.5 and 31 m

## 3.4 Signal preservation in the Adamello ice core

Evidence from the Adamello ice core has shown that proxies in temperate glacier must be approached with caution. Festi et al. (2021) observed that peaks in pollen concentration and black carbon concentration, with exceptional high values, may represent multiple years condensed in one single layer as a result of negative mass balance years and enrichment of the impurities at the exposed surface. This poses an issue for annual layer counting. Despite this, due to the alignment of many palynomorphs and black carbon peaks, the seasonality of the signal seems to be preserved. On the contrary, in Festi et al. (2021) the $^{210}$Pb

profile did not show a clear exponential decrease in activity concentrations with increasing depth, as it is typically observed in glacier ice. $^{210}$Pb concentrations have shown in literature large fluctuations connected with dirt horizons (Gaggeler et al., 1983) indicating that $^{210}$Pb may be transported with water, thus preventing a meaningful dating of temperate glaciers with this nuclide. $^{210}$Pb shows a behaviour in ice similar to what observed for cesium: these highly insoluble elements remain bound to particulate matter and are prone to relocation and loss of signal when meltwater is present. Unexpectedly, in this ice core

the cesium peak was not coincident with a dust layer. Furthermore, by comparing the highest $^{137}$Cs peak to activity levels attributed to 1963 found at Colle Gnifetti (Eichler et al., 2000), we observed a loss of expected signal of more than 50%. However, when examining tritium, the record appears to be mostly preserved, at least for the period post-1960s (as the 1954 and 1958 peaks were not detected in this ice core). This conclusion is drawn from the shape of the tritium profile matching the tritium precipitation record and from comparing the total inventory, where no loss in tritium activity is found. This provides

further confirmation of the better preservation of tritium compared to cesium. We thus believe that Adamello glacier can still

function as a paleoclimate archive. However, to correctly interpret the information thereby contained, it is necessary to consider melting and disturbance processes affecting the climatic signals.

## 4   Conclusions

We analyzed tritium with Liquid Scintillation Counting in the entire 2016 Adamello ice core with a resolution of approximately
0.5 m, and with an increased resolution for the portion between 27 and 35 m of depth. Our analysis showed contamination of tritium between 19 and 32 m of depth, attributed to the worldwide radioactive contamination caused by atmospheric nuclear bomb testing in the 1950s and 1960s. Additionally, we were able to compare the $^3$H profile with another byproduct of nuclear bomb testing, $^{137}$Cs. The analysis revealed that the radioactive peak associated with 1963 is not coincident for the two artificial radionuclides. The $^3$H peak occurs 1.5 m above the $^{137}$Cs one. As many records show that the deposition of the two radioactive
species was concurrent, the shift of the peaks is only explainable by assuming that the $^{137}$Cs peak has been subject to a downward relocation triggered by meltwater percolation in temperate ice. Tritium was less affected by the process as it is present in the ice matrix. On the contrary, $^{137}$Cs, being more soluble and mobile in aqueous environments, is more prone to relocation if liquid water is present. While the total tritium inventory associated with the 1963 peak in the Adamello ice core generally agrees with previous findings for cold glaciers not affected by melting, we find that this is not the case for 137Cs.
While the total tritium inventory associated with the 1963 peak in the Adamello ice core generally agrees with previous findings for cold glaciers not affected by melting, we find that this is not the case for $^{137}$Cs. The total inventory of $^{137}$Cs calculated at the 1963 peak for this radionuclide is among the lowest ever reported, providing additional evidence that the position of cesium within the ice was disrupted by meltwater, leading also to partial removal through washing. The results in this study indicate that there are potential issues with the established application of radionuclides for dating mountain ice cores. While for a cold
glacier it is reasonable to assume that radioactive signals are well preserved in the ice column, for temperate ice cores this is not the case. To avoid potential dating inaccuracies, it is preferable to reconstruct the entire profile of $^3$H to obtain a stronger indication of the true 1963 signal depth. In general, when disturbances are present in the reconstructed profile, matrix signals such as tritium are more easily preserved. Additional studies are needed to investigate the relocation mechanism of cesium in the presence of meltwater.

*Author contributions.*  ED and MC conceived the idea of this work. ED carried out the sample preparation for liquid scintillation counting. DB and RG acquired and analyzed the Hypersectral Data. ED, GB and BD intepreted the data. ED prepared the manuscript with contributions from all co-authors. MS and VM provided additional editing and review.

*Competing interests.*  The authors declare that they have no conflict of interest

*Acknowledgements.* This work is a contribution to the project CALICE – Calibrating Biodiversity in Glacier Ice, a multidisciplinary program between the University of Innsbruck, the Free University of Bozen-Bolzano, and the Fondazione Edmund Mach in San Michele, funded by the EVTZ/Austrian Science Fund (IPN 57- B22). This is CALICE project publication no. 2. We would like to thank all members of the CALICE scientific consortium, especially those who helped during the coring activities and the processing of the ice core. We are grateful to the ENEA drilling team and the Alpine guide Nicola Viotti (Guide Alpine Valsusa) for their excellent work during the coring campaign. Drilling has been possible thanks to a specific grant (POLLice) to FEM (Fondazione Edmund Mach) from the Autonomous Province of Trento (PAT) and logistic support (helicopter flights) provided by Ernesto Sanutuliana. EuroCold Lab activities were partially funded by the Italian Regional Affair Ministry.

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
