# Peer review of "Temporal markers in a temperate ice core: insights from 3H and 137Cs profiles in the Adamello glacier"

_The Cryosphere, 2023_

## Referee Comment (RC2)

Review of "Evidence of radionuclide fractionation due to meltwater percolation in a temperate glacier" by Elena Di Stefano, Giovanni Baccolo, Massimiliano Clemenza, Barbara Delmonte, Deborah Fiorini, Roberto Garzonio, Margit Schwikowski, and Valter Maggi.

The manuscript of Di Stefano et al., presents 3H and 137 Cs ice core profiles from the temperate Adamello glacier (3100 m asl, Italian Alps). Both parameters show enhanced concentrations which were attributed to reflect the effect of atmospheric nuclear bomb tests in the 1950s and 1960s. However, the records of the well-dated maximum of these bomb tests which is expected in snow in 1963 for both parameters, is displaced between the 3H and 137Cs depth profiles by 1.5 m.

There are two former publications from this Adamello ice core, one (Di Stefano et al., 2019) presenting the 137Cs depth profile and concluding that 137Cs is tightly bound to insoluble particulate matter inside the ice core, and one (Festi et al., 2021) which established a timescale in annual resolution for the ice core on the basis of black carbon and pollen seasonality in combination with radionuclides 210 Pb and 137 Cs.

The authors of this study here attributed the observed displacement of the 3H and 137Cs bomb test signals to post depositional melting processes which affect 137Cs more than 3H as it was observed in former studies in temperate glaciers from the Alps and from Svalbard. However, beyond this conclusion on the depth difference of the 3H and 137Cs 1963 peak in a temperate glacier, which is already known in literature, no further implications were outlined concerning e.g. the established timescale of this ice core, or on the general suitability to use this glacier site as paleo-archive. Therefore the manuscript does not fulfill the standard of The Cryosphere.

In addition, the discussion about the misalignment of 3H and 137Cs seems to be not conducted thorough. E.g., two 3H depth profiles are presented in the manuscript, one measured on the drilling chips and one on ice core samples, whereas the absolute 3H maxima of both data sets show a depth difference of 1.5m. Further 1.5m below is located the 137Cs peak, and again 1m deeper in the core there is an outstanding mineral dust layer. Comparing these depth displacements with the regular occurring black carbon and pollen horizons and the out of this resulting ice core dating (Festi et al., 2021) based on regular seasonal occurrences of pollen and black carbon peaks during summer it seems that the time interval between the two 3H and the 137Cs peaks, and the mineral dust layer are in the order of magnitude of 1 year each. Thus, the outstanding dust layer would correspond to accumulations from precipitation around 1959-1960, and should lie therefore still lie completely within the time interval influenced by the bomb tests and the resulting enhanced H3 and 137Cs contamination, as it was observed in European precipitations and deposition archives. However, in the Adamello core 137Cs as well as 3H showed pre bomb test levels at the corresponding ice core depth of this outstanding mineral dust layer, what questions whether the dating is wrong or the complete 3H and 137Cs bomb inventories before 1963 (or 1960?) were removed completely by percolation. And if percolation occurred, why was the 137Cs not at least partly accumulated together with the mineral dust in the outstanding dust layer below?

My recommendation would be that the authors revisit their data concerted with all of the already existing glacio-chemical and dating information to present a carful discussion of novel observations to draw a consistent picture of ice core signal preservation conditions at the site.

---

## Author Comment (AC1)

Response from authors for RC1 for manuscript "Evidence of radionuclide fractionation due to meltwater percolation in a temperate glacier"

Regarding the discussion about why we find a dust-rich layer approximately 1 meter below the misaligned 137Cs peak, our interpretation is that the notable dust peak below the 137Cs rich layer is indipendent and has not been influenced by 137Cs percolation. It is difficult to provide hypothesis circa the conservation signal of the dust peak and its origin without having the complete dust profile of the ice core. In regards to why the 137Cs was not at least partly accumulated in the dust-rich layer this is a problem we cannot solve definitely with current data. Nonetheless we think that the key information contained in our data series is that in glaciers were heavy amount of melting is expected under current climate conditions, the 137Cs data is not to be considered as reliable as it always has been up to now in the ice core dating community. We will add some text to the article to further clarify this point:

*"Dislocation of cesium particles inside the ice cores has undoubtedly happened, and the fact that a dust-rich layer with no cesium was found just below the cesium peak may be an indication that even a small amount of particulate (small enough to not be identified as a dust layer in the stratigraphy) can lead to a high cesium peak. On the other hand, a signal such as the tritium one which is a matrix signal, is more likely to either be conserved or be lost to melting, but we do not expect to observe relocation of the peak, and therefore we wish to recommend tritium analysis rather than cesium when a strong tie point is needed for datation."*

In regards to misalignment between the two tritium peaks (high and low resolution), we consider the high-resolution series to be more reliable since the low-resolution series provides a mean value over a thicker portion of ice (as the samples are usually 50-60cm instead of 5-10 of the high-resolution series) and the high-resolution series has shown notable variations over limited ice thickness. Nonetheless the discrepancy between the activity values is inside the error bars as shown in Figure 4 and both series are not compatible with the depth of the 137Cs signal. It cannot be possible to have a relocation of the tritium signal following the same mechanism proposed for cesium because of the very nature of this signal, which is tied to the water molecule itself and not to particulate matter. No traces of refreezing at this depth in the ice core samples were found, thus if meltwater was present, it percolated down to lower levels leading eventually to signal loss of tritium but not to mislocation of the peak.

---

## Author Comment (AC2)

Response from authors for RC2 for manuscript "Evidence of radionuclide fractionation due to meltwater percolation in a temperate glacier"

To our knowledge, the shift between the 3H and 137Cs 1963 peak has been reported in a single study from Pinglot et al. (2003), more than 20 years ago and in a completely different context from the Alps. Our study is the first documentation of this phenomenon for an Alpine mountain glacier presenting fully temperate conditions. We think that despite the early single study, it is important to acknowledge the discrepancies we found between the radioactivity profile of the two considered radionuclides. It is common to find in literature chronological markers related to 137Cs or 3H, without really discussing the profiles or comparing them. The ice core community that works on mountain glaciers always assumes that those proxies are perfectly preserved in glacier ice, but, as we highlight, this is not the case, in particular at the time of climate change and global warming. Those signals can no longer be considered as fully reliable. We believe that this is an important message to share with glaciological community and we also believe that the scopes of this journal perfectly align with this.

Regarding the established timescale we understand that additional comments may be needed and will add a paragraph to text as follows:

"*The established timescale may be slightly inaccurate regarding the layer counting between the 1986 and 1963 tie points, as it heavily relies on the implication that the 137Cs peak corresponded to the 1963 unaltered signal. The inaccuracy would introduce an additional error of 1 /2 years as the downward migration of the 137Cs signal is of only a couple of meters.*"

We will also provide in the text an additional paragraph on the suitability of this glacier as paleoarchive by adding a summary of all glaciochemical published data available for this ice core, as pointed out by the reviewer:

"*In Festi et al (2021) it is observed that peaks in pollen concentration and black carbon concentration with exceptional high values may represent multiple years condensed in one single layer as a result of negative mass balance years and enrichment of the impurities at the exposed surface. This poses an issue for annual layer counting. Despite this, due to the alignment of many palynomorphs and BC peaks, the seasonality of the signal seems to be preserved. On the contrary, in Festi et al (2021) the 210Pb profile did not show a clear exponential decrease in activity concentrations with increasing depth, as it is typically observed in glacier ice. 210Pb concentrations have shown in literature large fluctuations connected with dirt horizons (Gaggeler et al 1983) indicating that 210Pb may be transported with water, thus preventing a meaningful dating of temperate glaciers with this nuclide. 210Pb thus shows a behaviour in ice similar to what observed for cesium: these highly unsoluble elements remain bound to particulate matter and are prone to relocation and loss of signal when meltwater is present. Looking at 137Cs, by taking into account the highest 137Cs peak and comparing it to activity levels attributed to 1963 found at Colle Gnifetti (Eichler et al 2000), we found a loss of expected signal of more than 50%. On the other hand, looking at tritium the record seems to be mostly preserved at least for the period post 1960s (as the 1954 and 1958 peaks were not detected in this ice core), because the shape of the tritium profile matches the tritium precipitation record and because comparing the total inventory, no loss in tritium activity is found; this is further confirmation about the better preservation of tritium compared to 137Cs. We thus believe that Adamello glacier can still function as a paleoclimate archive but to correctly interpret the information thereby contained, it is necessary to consider melting and disturbances processes affecting the climatic signals*".

Regarding the discussion about the misalignment between the peaks, the two tritium profiles (high and low resolution) have a difference of 1.5m but looking at activity values in the different samples between 29 and 31m it can be seen that they are well within the error bars; thus we can only say with no doubt that the 1963 deposition lies in this interval but we cannot pinpoint it without additional markers. Considering this, the displacement of 3H and 137Cs peaks lies in a range between 1.5 and 3 meters. Considering that the dating reported in Festi et al (2021) heavily relies on the implication that the 137Cs peak corresponds to 1963, it is difficult to say if the layer rich in mineral dust actually corresponds to 1959-1960, as the error on the dating would be much larger extending the possible dating range to the early 1950s. It is true that the Adamello ice core does not display radioactivity peaks that could be attributed to the pre 1963 era such as the 1954 and the 1958 peaks often reported in literature; the possibility of removal by percolation, especially since activities were much lower than 1963 and that additional time has passed (and therefore decay), leading to a much more faint signal which could easily be lost in presence of massive meltwater.

In regards to why the 137Cs was not at least partly accumulated in the dust-rich layer, this is a problem we cannot solve definitely with current data. Nonetheless we think that the key information contained in our data series is that in glaciers where heavy amount of melting is expected under current climate conditions, the 137Cs data is not to be considered as reliable as it always has been up to now in the ice core dating community. We will add some text to the article to further clarify this point:

*"Dislocation of cesium particles inside the ice cores has undoubtedly happened, and the fact that a dust-rich layer with no cesium was found just below the cesium peak may be an indication that even a small amount of particulate (small enough to not be identified as a dust layer in the stratigraphy) can lead to a high cesium peak. On the other hand, a signal such as the tritium one which is a matrix signal, is more likely to either be conserved or be lost to melting, but we do not expect to observe relocation of the peak, and therefore we wish to recommend tritium analysis rather than cesium when a strong tie point is needed for datation."*

---

## Author Response (AR2)

Point by Point Reply for Reviewer 2 and modifications to the text

- Reviewer comment:
  My main concern is that the depth co-registration between chip and ice samples is robust enough to justify the conclusions. The manuscript focuses on three datasets: tritium in the ice, tritium in the chips, and 137Cs in the chips. Based on Festi et al. 2021, drilling was halted because of wet drilling conditions. Could such conditions compromise samples obtained from the chips or the depth registration of the chip samples relative to the ice? I can imagine if the borehole was waterlogged, chips could be mixed across a wide range of depths. Furthermore, if the borehole was waterlogged, was it at all possible to separate water from the chips, and if not, could such water contaminate or dilute data developed from the chip samples? These details, to me, are crucial to ensure that ice and chip measurements are co-registered in depth since the conclusions of the paper are based on the offset in 137Cs and tritium. If the 137Cs and tritium measurements were made on the same exact samples collected as chips, this uncertainty could be somewhat ameliorated. Is that this case? If so, that should be clearly stated.

Reply:

The ice core was extracted during the winter season, mainly working at night, to maintain a low temperature during the drilling activities done by an electromechanical system. There was no water in the borehole, if not the drilling system would not work, and the ice core was recovered dry. The chips were collected by recovery of the cutting in a specific close chamber above the core barrel. The external chips that were not in the chamber were rejected. For these reasons, we can reasonably exclude contamination of cutting from other parts of the hole. The conditions to which Festi et al 2021 refer were only found at the bottom of the ice core, and since this drilling system cannot work if water is present, that is the reason why only 46m of ice core were recovered.

Additionally, cesium and tritium measurements on the chips were indeed done on the exact same samples. The sample was filtered and the filter retaining the particulate matter was used for cesium analysis while the remaining filtered water was used for tritium analysis.

To better clarify these points we added the following text:

At line 90: "The chips were collected in a specific close chamber above the core barrel, which was emptied after the collection of each core sample."

At line 270: "The low resolution tritium dataset was prepared from the same exact samples which were also used for  137Cs analysis."

- Reviewer comment:
  I did not think the linkages to the SDI dataset were convincing or added much to the manuscript. It seemed extraneous unless the authors could draw a stronger link/conclusion

based on it. They also do not mention the smaller SDI peak that aligns with the 137Cs peak (though this does not consider depth registration uncertainties between the chip and ice measurements).

Reply:

The SDI dataset was added as it was the only other available information on particulate matter we had on this ice core (besides the Black Carbon dataset published by Festi et al 2021). We think that the fact that no big peak is present in the SDI dataset at the same depth as the cesium main peak indicates that the Cesium peak is not due to an abundance of dust in this layer, and this should be important information since the Cs data is not calibrated on the amount of dust present but on the volume of filtered water, despite Cs being in particulate form. (This was not done as it was not possible for us to quantify the amount of dust deposited on the filter used to prepare the cesium samples.) This is valid under the assumption that the depth registration of the chips is correct.

- Reviewer comment:
  Lastly, I do not think that the author's conclusion that tritium is not impacted by downward relocation is justified without additional independent dating. The authors are only assuming that tritium is not impacted since it is shallower in the ice than Cs137. Such independent dating may be impossible, but just because the tritium is above the 137Ce peak does not mean it has been affected by melt. I'd simply suggest removing this conclusion.

Reply:

We agree with the reviewer and we did not intend to imply that tritium is not impacted by meltwater, but that the impact that meltwater can potentially have on the two radionuclides is different. If relocation of meltwater to adjacent layers is taking place, tritium is more likely to show a broader signal as opposed to a well-defined and constrained peak. Additionally, signal loss may be present if meltwater containing tritium escapes the system. We do not think it is likely to have a shift of the peak if no other evidence of massive melting is present (e.g. lack of bubbles). To better convey the point and avoid misunderstandings we changed the text at line 253 from "Tritium was not affected by the process as it is present in the ice matrix" to "Tritium was less affected by the process as it is present in the ice matrix."

---

## Author Response (AR3)

We wish to thank the editor for the time dedicated to the revision of this manuscript and for the comments provided. All of the comments were integrated into the text, captions, and figures, as can be seen by the marked-up version of the manuscript. Here you will find a point by point description of the revisions made, with the editor's comments marked in blue and our response marked in black.

Abstract: 'meltwater disturbed its distribution' does not really make sense -> please try to revise so that it's more clear what process you want to emphasize here. Is it that the meltwater distribution within the the ice is more uniform?

This was changed to "the presence of meltwater affected the distribution of this radionuclide inside the ice" to make it more clear.

Figure 1: the choice of the color for the lines is very similar: can you make one color red and the other black or dark green? Please consult with the color scheme recommended for scientific papers, for example:
https://www.simplifiedsciencepublishing.com/resources/best-color-palettes-for-scientific-figures-and-data-visualizations
Also , make with width of both lines a bit ticker. The panels have lots of white space, which is somewhat suboptimal for the size of the figure.
Also, the standard of x-y plots with just one y-axis to have the y-axis on the left. Please correct this. Make sure to include any units in the brackets.
Add a label to x-axis, for example: Time (year)'. All axis need to have labels.

The colors were changed to red and dark green,the width of the lines was made thicker, and the y-axis was moved to the left side. The x-axis labels were changed adding "Time (Years)" and tilting the single labels, as the image was made smaller to fit into one column (width 8.3cm as suggested by the Copernicus latex format).

Figure 2: Make it smaller as the figures has too much white space and it's unnecessarily wide. Make the new figure so that it can fit one column when published (I think that is 7.5 cm in width). You can enlarge the markers for the data points in the plot. The figure capture does not provide sufficient information: state in the captions the full name of CPS and Bq/L, and include their units if there are not unit-less. Units should be written within brackets. Also, provide more information on what you included in the figure: for example, coefficient of determination ($R^2$) and the linear relationship between CPS and Bq/L as obtained by a linear regression model or a linear fit.
Also, you state: 'Errors on cps are 0.001'. Do you provide more details in the text on how these errors are obtained. If not, please do so. If yes, refer in the figure caption to that information in the text, for example: 'Errors calculated from XXX (see the main text) on CPS are 0.001.' Please note that some calculation cannot just randomly appear in the figure only, without it being explained or referred to in the main text.

This figure was also made smaller (8.3cm width) while enlarging the datapoints and thickening the lines. The y-axis label was changed to "Count rate (cps)". The caption was changed to: "Calibration curves calculated for the 3 three sets of tritium standards Data points are plotted as counts per second (cps) vs activity values (Bq/ml). For each set the dashed line represents the linear fit

between the data points, which is also reported in the left corner of the image together with the coefficient of determination (R2). Errors on the count rate, calculated as the square root of each measurement divided by the count time, are 0.001 cps."

General comments for the figure captions and main text: when you report a number smaller or equal to 10, spell the number our, so should be 'three samples' instead of '3 samples'. For numbers larger than ten, you should type them as numbers.

This was changed throughout the text and captions.

Figure 3: Caption: please make the full sentences, for example: Low resolution measurements referring to ice chips are shown in black dots, the observed activities and their error bars are shown in the histogram and their respective bars, etc.

The caption was changed to: "Tritium activity in the Adamello temperate ice core. Low-resolution measurements referring to ice chips are shown as black histograms where the width of each bar represents the actual depth covered by the sample and the height represents the Specific Activity (Bq/ml). Error bars indicate the uncertainty on the Specific Activity, calculated following the propagation of error method. Diamonds represent samples from the low-resolution dataset for which the Upper Limit was calculated instead of the Specific Activity, as detailed in the main text. High resolution measurements referring to ice core samples are shown in red."

General comments for the figure captions: Provide enough information in the figure captions so that they can be read as a stand alone, i.e. the reader does not need to dig into the text to figure out what you are presenting in the figure. Any abbreviations need to be restated.

Figure 4: The line widths, especially in panel (b) are too thin and barely visible -please enlarge the width where possible. Also, please increase the font size in all legends, titles, and axis labels. Make sure to restate what each abbreviation represents (e.g. SDI).

Line width were enlarged in all four panels, as the font size. The two panels on the right side were made smaller to reduce the white space. The caption was changed to: "Detailed overview of the ice core in the portion where the the tritium and 137 Cs 1963 peaks have been observed (between 25 and 35 m deep). a) Pollen and spores concentration taken from Festi et al. (2021). Each peak of pollen and spores concentration reflects one flowering year (February–September). b) Tritium profiles. Specific activity (Bq/L ) of 3H for the ice chip low-resolution dataset is shown as black histograms while the ice core high-resolution dataset is shown in red. Error bars indicate the uncertainty on the Specific Activity, calculated following the propagation of error method. The diamonds indicate upper limits calculated for the high-resolution data set, as detailed in the main text. c)Snow Darkening Index (SDI) record, a hyperspectral index related to the impurity content of ice. d) Specific activity (Bq/L ) of 137 Cs taken from Di Stefano et al. (2019)"

Figure 5: Again, y-axis should be shown on the left side of the graph. The panels have way too much white space, consider downsizing the figure The figure caption does not explain what the histogram represents (distribution?) and what the bars represent (error?). Make sure to restate what the abbreviations are and include all units.

Y-axis was moved to the left side and the entire figure was downsized to minimize white space. The caption was changed to: "The upper panel shows the total beta activity from the Lys ice core. Total beta values are reported here as count rate in counts per minute (cpm) with the associated uncertainty. The lower panel shows the Specific Activity (Bq/kg) of 137 Cs with the associated error, as reported in Clemenza et al. (2012). In both panels the width of each bar represents the actual depth covered by each sample."

Comments regarding you response to the reviewer:
You responded to the comments about the linkages to the SDI dataset, but did not make any revisions in the text. I suggest you address this issue in the text and state more clearly what the motivation was for the inclusion of the SDI data in the text (something along the lines you wrote in the response: no big peak is present in the SDI dataset at the same depth as the cesium main peak indicates that the Cesium peak is not due to an abundance of dust in this layer, and this should be important information since the Cs data is not calibrated on the amount of dust present but on the volume of filtered water, despite Cs being in particulate form).

We added a paragraph at line 201 to address this issue: "the absence of any significant peak in the SDI dataset at the same depth as the main cesium peak indicates that the cesium peak is not caused by an abundance of dust in this layer. Since the impurity and the radioactive layer don't overlap, we ruled out this possibility. This finding is significant because the cesium data is not calibrated on the amount of dust present but on the volume of filtered water, despite Cs being in particulate form. Moreover, the absence of a dust-rich layer at 32m of depth suggests that even a minimal amount of particulate matter can contribute to a substantial cesium peak."

Line 257: Revise to: 'While the total tritium inventory associated with the 1963 peak in the Adamello ice core generally agrees with previous findings for cold glaciers not affected by melting, we find that this is not the case for 137Cs.

This was revised as suggested.

Line 260: 'that meltwater disturbed its distribution' – as I pointed already above this phrase does not make sense.

The line was changed for more clarity to: "providing additional evidence that the position of cesium within the ice was disrupted by meltwater, leading also to partial removal through washing."

Line 2601: 'Such finding' -> not clear to what finding you are referring to as this is a new paragraph. Need to restate in order to make this more clear. For example: 'The results in this study indicate that there are possible issues ...'
Change to: 'for a cold glacier'

This was revised as suggested.